# A Comparative Evaluation of HbA1c Measurement Methods and Their Implications for Diabetes Management

**DOI:** 10.3390/diagnostics13223449

**Published:** 2023-11-15

**Authors:** Hyeokjun Yun, Joo won Park, Jae Kyung Kim

**Affiliations:** 1Department of Medical Laser, Graduate School of Medicine, Dankook University, Cheonan 31116, Republic of Korea; 10621yhj@naver.com; 2Department of Laboratory Medicine, College of Medicine, Dankook University, Cheonan 31116, Republic of Korea; joowon@dankook.ac.kr; 3Department of Biomedical Laboratory Science, College of Health & Welfare, Dankook University, Cheonan 31116, Republic of Korea

**Keywords:** correlation coefficient, HbA1c, performance evaluation, point-of-care testing, rapid diagnostic kit

## Abstract

In this study, we assessed the correlations between hemoglobin A1c (HbA1c) measurements obtained using three different diagnostic methods, namely reversed-phase cation-exchange chromatography, high-performance liquid chromatography, and lateral flow immunoassay (LIFA) with an AnyLab F instrument. HbA1c levels measured with the AnyLab F instrument and those measured with the HA8190V, HA8180, and D100 instruments were strongly correlated. High R-square values and low *p*-values indicated significant and reliable correlations, supporting the clinical interchangeability of these methods. Notably, demographic and clinical analyses revealed uniform HbA1c levels across age groups, suggesting minimal age-related variations in HbA1c levels in the cohort. This finding has implications for diabetes management strategies across different age groups, emphasizing the versatility of the AnyLab F instrument. Overall an average HbA1c level of 7.857% among diabetes mellitus-diagnosed participants suggests moderately elevated HbA1c levels, underscoring the need for improved diabetes management. Younger individuals exhibited lower HbA1c levels, potentially owing to heightened awareness and treatment plan adherence. Conversely, older adults had higher HbA1c levels, likely influenced by age-related changes and comorbidities. Larger sample sizes and a comprehensive evaluation of various measurement principles are needed to strengthen the findings herein. Additionally, exploring additional biomarkers and assessing LIFA performance in larger sample sets will advance the clinical utility of HbA1c measurements.

## 1. Introduction

The prevalence of diabetes mellitus (DM) is increasing worldwide, and the World Health Organization estimates that the global DM population will reach approximately 590 million by 2035 [1]. Socioeconomic development has resulted in lifestyle changes, including eating habits, which have in turn increased the number of patients with DM worldwide. Furthermore, with the aging of the global population, the frequency of chronic conditions, such as DM and metabolic syndrome, is expected to rise even further. Chronic complications, including microvascular complications, often manifest before the onset of DM, making an early diagnosis crucial for prevention. To achieve this, it would be advantageous to develop a simple and accurate diagnostic test that does not require fasting, as conventional methods of fasting blood glucose or oral glucose tolerance testing can be inconvenient [2].

Hemoglobin A1c (HbA1c) is a form of hemoglobin that comprises two alpha chains and two beta chains, with glucose or glucose derivatives non-enzymatically bound to the hemoglobin molecule. Including HbA1c level measurement as a diagnostic criterion for DM eliminates the need for fasting, providing stable results even in patients following 6–8 weeks of lifestyle changes [3]. The measurement of HbA1c levels can also function as an indicator for chronic complications linked to DM. An HbA1c level of >6.5% is regarded as an indicator of a high risk of DM. HbA1c testing has thus been proposed as a useful strategy for evaluating a patient’s overall glycemic control in various studies, including the Diabetes Control and Complications Trial, completed in 1993 [3,4]. Furthermore, strict glycemic control achieved through HbA1c level monitoring can substantially reduce the progression of serious DM complications, such as DM retinopathy and microvascular diseases. The intra-individual variation in HbA1c levels is <2%, which is more stable than the 12–15% variation in fasting blood glucose levels, making it a promising diagnostic criterion for DM [5]. However, more than 30 different HbA1c measurement devices based on various methods have been developed, meaning that standardization is now necessary.

The National Glycohemoglobin Standardization Program (NGSP) and the International Federation of Clinical Chemistry and Laboratory Medicine (IFCC) in the United States (U.S.) are leading programs aiming to achieve HbA1c standardization [4]. In the NGSP standard measurement method, ion-exchange high-performance liquid chromatography (HPLC) is conducted to measure the area under the curve of the HbA1c peak, whereas the IFCC standard method is used to measure the levels of N-terminal hexapeptide generated by treating the sample with endoproteinase Glu-C using HPLC–mass spectrometry or HPLC–capillary electrophoresis [1,2,3,6]. In 2009, the International Expert Committee under the International Diabetes Federation proposed an HbA1c level of ≥6.5% as a diagnostic criterion for DM; this limit was subsequently adopted by the American Diabetes Association as an additional criterion for the diagnosis of DM. However, it is important to note that these criteria rely on the standardization of HbA1c measurements using NGSP-approved methods [6].

In LIFA, monoclonal or polyclonal antibodies are used as bioreceptors on nitrocellulose membranes for protein detection [5,7,8]. LIFA is economical and easy to use, and it can be manually performed within 15 min. This assay overcomes the errors of conventional LIFA, the results of which are interpreted qualitatively by the naked eye, resulting in errors due to inadequate visual sensitivity [9,10]. To increase the clinical usefulness of HbA1c levels, rapid, accurate, and convenient inspection assays, including LIFA, are continuously being developed and evaluated. In the present study, we focused on the AnyLab F system, a recently developed measurement device based on LIFA.

LIFA is an emerging and increasingly significant technology in the realm of medical diagnostics [11]. In this context, the primary focus of this study was to apply LIFA to measure HbA1c levels and contrast it with conventional techniques such as reversed-phase cation-exchange chromatography and HPLC. The underlying motivation for employing LIFA in this study is rooted in the demand for diagnostic methods that are more accessible, cost-effective, and precise, particularly for monitoring conditions like diabetes [11,12].

LIFA is generally regarded as cost-effective compared to more advanced techniques like HPLC. The cost of materials and reagents for LIFA is relatively lower, making it a more budget-friendly option for diagnostic applications. On the other hand, HPLC methods typically entail higher costs due to the requirement for specialized equipment, high-quality reagents, and skilled personnel. Maintenance costs for equipment can also be substantial [5,7,11].

Regarding infrastructure needs, LIFA typically does not necessitate complex infrastructure. It can be conducted in various settings, including clinical laboratories, doctor’s offices, or even in the field. Basic laboratory facilities and a clean working environment are generally sufficient. In contrast, HPLC methods demand sophisticated laboratory setups with specialized equipment, such as chromatography systems and high-quality detectors. Controlled laboratory conditions, including temperature and humidity control, are essential for ensuring accurate results [9,11].

Lateral flow immunoassays are relatively straightforward to carry out and interpret. They do not require highly specialized personnel and can be performed by trained technicians or healthcare professionals. However, HPLC techniques require skilled personnel with expertise in operating and maintaining sophisticated equipment. Analyzing the data obtained from these methods may also require the involvement of experienced analysts or scientists with a background in chromatography and related techniques [10,12].

Considering these factors, LIFA appears to have advantages in terms of cost-effectiveness, rapid results, and ease of use compared to the more elaborate conventional techniques, although there are some reduction in analytical precision and sensitivity.

Comparative studies are indispensable for determining the effectiveness and dependability of LIFA, particularly when juxtaposed with traditional methodologies. This research carries substantial importance in the sphere of medical diagnostics owing to the potential for innovation and enhanced patient care. Furthermore, this correlation study holds significance as it evaluates a novel diagnostic technology, the AnyLab F system, and gauges its performance relative to that of established approaches. The potential advantages, such as improved diagnostic precision, resource efficiency, and superior patient care, underscore the value of this study for healthcare professionals, researchers, and patients [13,14].

We aimed to evaluate and confirm the utility of AnyLab F by comparing its performance with that of three other measurement devices, namely ARKRAY ADAMS HA-8180 (ARKRAY KDK, Kyoto, Japan), which measures HbA1c levels using reversed-phase cation-exchange chromatography, as well as D-100 (Bio-Rad Laboratories, Hercules, CA, USA) and ADAMS™A1c HA-8190V (ARKRAY, Inc., Kyoto, Japan), both of which measure HbA1c levels using the HPLC method [2,3,4,15].

## 2. Materials and Methods

### 2.1. Study Design and Participants

Between August and September 2022, HbA1c measurements were conducted on blood samples collected from both DM (122) and non-DM (78) individuals at Dankook University Hospital (Cheonan, Republic of Korea). All participants visited the hospital and underwent standard biochemical tests specifically designed to measure HbA1c levels. This study was approved by the Clinical Research Review Committee of Dankook University (Institutional Review Board DKU Certificate No. 2022-10-030). Retrospective data were obtained, excluding personal information, and the approval was processed from Dankook University Hospital through a consent form and an exemption from review (Figure 1).

### 2.2. Data Collection

HbA1c levels in whole blood were measured using AnyLab F1, an immunofluorescent quantitative analyzer developed by Z-Biotech (Cheongju, Republic of Korea). The Diagnostic Test Department at Dankook University performed all tests in accordance with the manufacturer’s testing procedures. Quantitative data were obtained from whole-blood samples of 200 participants, including 122 patients with DM.

HbA1c measurements were also performed using other instruments: ARKRAY ADAMS HA-8180 (ARKRAY KDK, Kyoto, Japan), D-100 (Bio-Rad Laboratories, Hercules, CA, USA), and ADAMS™A1c HA-8190V (ARKRAY, Inc., Kyoto, Japan). The Diagnostic Test Department at Dankook University assessed the data following the testing procedures specified for each measurement instrument. HbA1c levels were further obtained from 200 whole-blood samples.

The retention of data and samples was governed by institutional policies and legal requirements. The disposal of samples and data followed approved protocols.

### 2.3. Immunofluorescence-Based Quantitative Assay

The method we conducted uses a sandwich immunodetection principle, in which a fluorescence-labeled detector antibody specific to HbA1c is attached to the target protein within the sample [14]. Subsequently, the sample is applied to a test strip, and a secondary antibody embedded in the solid phase captures the fluorescence-labeled antigen–antibody complex. The intensity of the fluorescence signal produced by the captured complex directly correlates with the amount of antigen present, allowing for the determination of the sample antigen concentration through a predefined calibration process.

To initiate the process, the detection buffer and serum are combined, causing the antibody within the detection buffer and antigen within the specimen to form an antigen–antibody complex. This mixture is then dispensed into the sample well of the cartridge, where it binds to the antibody coated on the nitrocellulose membrane and triggers a sandwich immune response. The magnitude of the immune response within the sandwich structure is subsequently translated into a fluorescent signal, and the concentration can be determined using the specialized AnyLab F1 measurement device.

### 2.4. LIFA Quantitative Measurement

The AnyLab F point-of-care testing system was designed for the quantitative measurement of DM markers, including HbA1c levels in whole blood, based on the principle of LIFA. The test results are presented as percentages (%). To ensure quality control, a fluorescence-labeled control protein was included in the reaction, and the intensity of the control line was measured.

All the necessary components for the assay, including the reaction components and meter, were provided by the manufacturer. The assay was performed according to the manufacturer’s instructions. Briefly, 10 µL of whole blood was mixed with a predetermined volume of detection buffer containing fluorescence-labeled anti-monoclonal antibodies and anti-rabbit IgG. A 100 μL aliquot was then loaded into the sample wells of each test strip within the cartridge, which was then incubated at 15–30 °C for 15 min. The intensity of the captured fluorescence-labeled antigen–antibody complexes were measured using a meter, and the level of HbA1c in the sample was determined. The accuracy and precision of the assay were determined using the internal quality control materials supplied by the manufacturer. After 15 min incubation at room temperature, the cartridge was inserted into the measurement device, and the quantitative results for each parameter were displayed on the AnyLab F1 screen (Figure 2).

### 2.5. Data Evaluation

The 200 samples used in this study were derived from excess de-identified serum samples originally submitted to the laboratory. The selected samples encompassed the analytical range of the AnyLab F assay, with HbA1c levels ranging from 4.0% to 15.0%.

### 2.6. Statistical Analyses

Normally distributed continuous data are expressed as the mean ± standard deviation. An independent *t*-test was used to determine the correlation among the multiple assays. All statistical analyses were performed using GraphPad Prism (version 7.00.159, Dotmatics). Statistical significance was set at *p* < 0.05.

## 3. Results

### 3.1. Correlation of HbA1c Levels Measured Using AnyLab F and HA8190V

As shown in Figure 3, we observed a strong correlation between the whole-blood HbA1c levels measured using the AnyLab F and HA8190V diagnostic methods. Within the 95% confidence interval (CI), the R^2^ value was 0.9617, indicating a strong correlation between the measurements (*p* < 0.0001). The high R-squared value of 0.9617 within the 95% confidence interval suggests a strong linear relationship between HbA1c measurements obtained from the AnyLab F and HA8190V methods. In other words, changes in the results from one method closely corresponded to changes in the results from the other method.

### 3.2. Correlation of HbA1c Levels Measured Using AnyLab F and HA8180

As can be seen in Figure 4, a significant correlation was observed between the whole-blood HbA1c levels measured using the AnyLab F and HA8180 diagnostic methods. Within the 95% CI, the slope was 0.9801, and the X and Y intercept values were −0.07456 and 0.07307, respectively. The R^2^ value was 0.97, indicating a strong positive correlation between the measurements (*p* < 0.0001). This finding has important implications for clinical practice, research, and patient care in the context of diabetes management as it underscores the reliability of these methods and their interchangeability for assessing HbA1c levels in whole blood.

### 3.3. Correlation of HbA1c Levels Measured Using AnyLab F and D100

As depicted in Figure 5, a substantial and statistically significant correlation was found between whole-blood HbA1c levels measured using the AnyLab F and those measured using the D100 diagnostic methods. Figure 4 illustrates a strong positive linear relationship between the HbA1c measurements obtained from the AnyLab F and D100 methods, with an R-squared value of 0.9679 within the 95% CI.

### 3.4. Demographic and Clinical Characteristics of Enrolled Participants

Following the confirmation of a strong correlation between AnyLab F and three other instruments, we conducted an analysis of the demographic and clinical characteristics of the study participants. The study population comprised 200 individuals, divided into two age groups: 100 young individuals (aged 0–59 years) and 100 older adults (aged ≥60 years). Notably, there was no statistically significant difference in HbA1c levels between the two age groups, with values of 6.992 ± 1.179% for the younger group and 7.296 ± 1.460% for the older group (Figure 6a). Within the study population, 127 participants were male, and 73 were female. Notably, there was no statistically significant difference in HbA1c levels between the male (7.172 ± 1.337%) and female (6.999 ± 1.338%) groups (Figure 6b). These findings underscore the absence of significant age- or sex-related disparities in HbA1c levels among the enrolled participants.

### 3.5. HbA1c Levels and Diabetes Control among Different Age Groups

In this study, 122 participants were identified as having DM based on criteria established by both the IFCC and NGSP, which define a diagnosis of DM as having HbA1c levels ≥6.5%. The overall average HbA1c level among these DM-diagnosed individuals with HbA1c levels ≥6.5% was found to be 7.857%, with a standard deviation of 1.188%. This indicates that, on average, these participants exhibit moderately elevated HbA1c levels, suggesting suboptimal blood sugar control (Figure 7a). Within this subgroup of 61 participants aged 0–59 years, the average HbA1c level was 7.575%, with a standard deviation of 1.049%. This implies that younger individuals with DM tend to have relatively lower and potentially better-controlled HbA1c levels than the overall DM group. The other subgroup, also consisting of 61 participants, comprised older adults aged ≥60 years. In this group, the average HbA1c level was higher at 8.139%, with a standard deviation of 1.259%. This suggests that older adults with DM, on average, exhibit higher and less well-controlled HbA1c levels than their younger counterparts (Figure 7b).

## 4. Discussion

We aimed to evaluate the correlation between the HbA1c measurements obtained using three different diagnostic methods, namely reversed-phase cation-exchange chromatography, HPLC, and LIFA using the AnyLab F instrument [1,16]. Our results revealed significant correlations between HbA1c levels measured using the AnyLab F instrument and those measured using the HA8190V, HA8180, and D100 instruments.

Slopes within the 95% CIs indicate the degree of agreement between the measurements [17]. These high values indicate a proportional relationship between the measurements obtained from AnyLab F and those obtained from the other instruments. Furthermore, the intercept values for the correlations provide insights into the origin of the data.

The high R^2^ values obtained for all three correlations indicate a strong relationship between the HbA1c measurements obtained from AnyLab F and those obtained from the other instruments. The R^2^ value represents the proportion of variation in the dependent variable (HbA1c), which can be explained by the independent variable (the measurement method). Therefore, a higher R^2^ value indicates a better fit between the measurements, suggesting a stronger correlation. The *p*-values for all three correlations were <0.0001, indicating statistical significance. This suggests that the observed correlations are unlikely to occur by chance and further supports the validity of the correlations we observed. Practically, this correlation may allow healthcare facilities or researchers to choose between the AnyLab F system and the other three instruments based on factors like availability, cost, or convenience, with the assurance of highly consistent results [17,18].

Overall, our findings demonstrated that HbA1c measurements obtained using the recently developed AnyLab F instrument were substantially correlated with those obtained using the HA8190V, HA8180, and D100 instruments. This strong correlation could have practical implications for healthcare settings, suggesting that either of the two methods could be used interchangeably to measure HbA1c levels, depending on availability or cost considerations, without compromising the accuracy of the results. LIFA offers advantages such as cost-effectiveness, ease of use, and rapid results [11,17]. Its quantitative feature overcomes the limitations of standard qualitative LIFA tests, thereby enhancing the accuracy and reliability of HbA1c measurements [13,19]. With the ongoing developments in LIFA technology and continuous evaluation of various instruments, the clinical utility of HbA1c measurements using LIFA can be further enhanced [12,19].

Upon analyzing the demographic and clinical characteristics of our study population, several noteworthy observations can be made. The classification of our participants into two age groups, young individuals (aged 0–59 years) and older adults (aged ≥60 years), revealed that there was no significant difference in HbA1c levels between these two groups. This finding suggests that HbA1c levels do not exhibit a substantial age-related variation within our study cohort [20,21]. Such uniformity in HbA1c levels across age groups may have implications for diabetes management and prevention strategies across the lifespan. Furthermore, these findings establish a baseline understanding of HbA1c levels within our study population, free from age- or sex-related bias [20,22,23]. This information can guide healthcare professionals in tailoring diabetes management approaches and interventions to more effectively cater to the specific needs of different patient groups [16,23,24]. Additionally, these findings emphasize the robustness of AnyLab F as a reliable instrument for measuring HbA1c levels, as it consistently correlated with other established methods across diverse demographic categories.

The findings of this study provide important insights into the relationship between HbA1c levels and diabetes control, particularly in different age groups. The overall average HbA1c level of 7.857% among DM-diagnosed participants suggests that, on average, these individuals have moderately elevated HbA1c levels. This indicates that the study population, as a whole, may be experiencing suboptimal blood sugar control [19,23]. Elevated HbA1c levels are associated with an increased risk of diabetes-related complications, highlighting the importance of improving diabetes management strategies [23,25]. The average HbA1c level of 7.575% in the young group suggests that younger individuals with DM tend to have relatively lower and potentially better-controlled HbA1c levels than the overall DM group. This could be due to factors such as increased awareness of diabetes management, adherence to treatment plans, and potentially more active lifestyles [21,26]. The higher average HbA1c level of 8.139% in the old age group implies that older adults with DM, on average, exhibit higher and less well-controlled HbA1c levels than their younger counterparts. This finding could be attributed to various factors, including age-related physiological changes, multiple comorbidities, and polypharmacy, which can complicate diabetes management [17,21,23,24]. While this study provides valuable insights, further research is needed to explore the factors contributing to the observed differences in HbA1c levels between age groups [20,23,27]. Additionally, investigating the impact of various treatment strategies on different age groups could inform more tailored approaches to diabetes management.

This study has some limitations. First, the sample size was relatively small (200 participants). Future studies with larger sample sizes are required to provide more robust evidence to indicate the observed correlations. Second, we compared the performance of the AnyLab F instrument with three other instruments. Additional comparisons with an HbA1c measurement device based on various other principles will contribute to a more comprehensive understanding of the performance and utility of AnyLab F [27,28,29]. Another limitation is the small number of serological biomarkers evaluated. Further studies should evaluate more metabolic syndrome-related biomarkers (e.g., thyroid-stimulating hormone and troponin I) to assess LIFA’s performance compared to that of HPLC or other methods, such as electrochemiluminescence immunoassay, to demonstrate its advanced effectiveness considering both cost and utility [18,25]. Furthermore, assessing LIFA’s performance in a larger number of samples would contribute to its development and improvement [18,30].

## 5. Conclusions

This study provides valuable insights into the measurement of HbA1c levels and its implications for diabetes management. The evaluation of three different diagnostic methods, namely reversed-phase cation-exchange chromatography, HPLC, and LIFA using the AnyLab F instrument, revealed significant correlations and interchangeability between these methods. The high R-square values and low *p*-values indicate strong relationships and statistical significance, supporting the validity of using AnyLab F in clinical practice. This finding has practical implications, allowing healthcare facilities and researchers to choose between these methods based on factors like availability, cost, or convenience, with confidence in the consistency of results.

Notably, our analysis of demographic and clinical characteristics showed that age-related variations in HbA1c levels within our study population were minimal. This uniformity in HbA1c levels across age groups suggests that age should not be a primary factor in diabetes management decisions. Furthermore, these findings underscore the robustness of the AnyLab F instrument, which consistently correlated with other established methods across diverse demographic categories.

However, it is essential to acknowledge the limitations of this study, including a relatively small sample size and a focus on a limited number of serological biomarkers. Future research with larger sample sizes and comprehensive evaluations of various measurement principles will provide a more robust understanding of HbA1c measurement methods and their clinical utility [18,30].

In conclusion, this study contributes to our understanding of HbA1c measurement methods, their comparability, and the role of age in diabetes management. These insights are vital for healthcare professionals in tailoring diabetes management approaches and interventions to more effectively cater to the specific needs of different patient groups. Ultimately, the findings herein are expected to enhance the accuracy of HbA1c measurements, improve diabetes care, and reduce the risk of diabetes-related complications for all individuals, regardless of age or demographic characteristics.

## Figures and Tables

**Figure 1 diagnostics-13-03449-f001:**
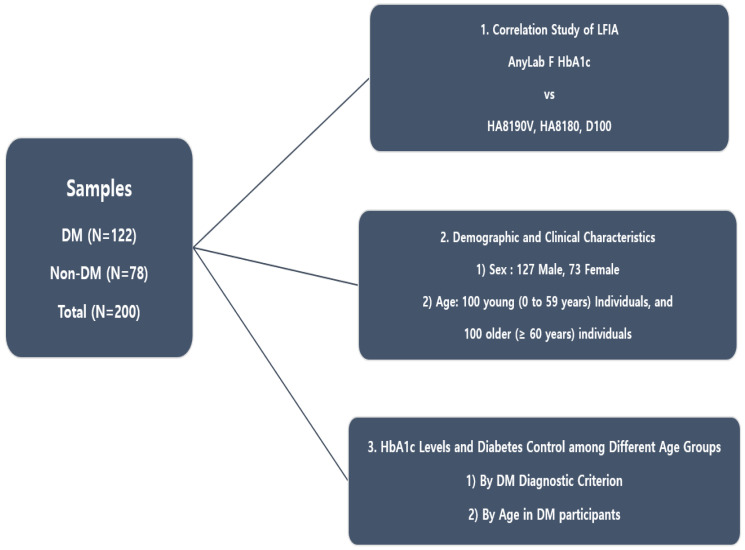
Flowchart of the experimental design.

**Figure 2 diagnostics-13-03449-f002:**
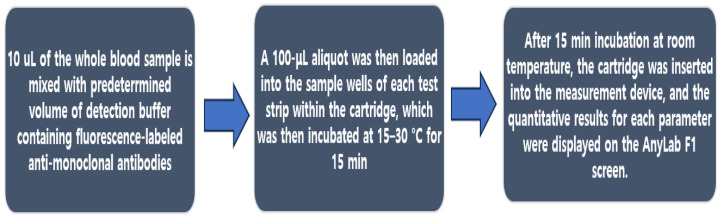
Flowchart of the LIFA protocol.

**Figure 3 diagnostics-13-03449-f003:**
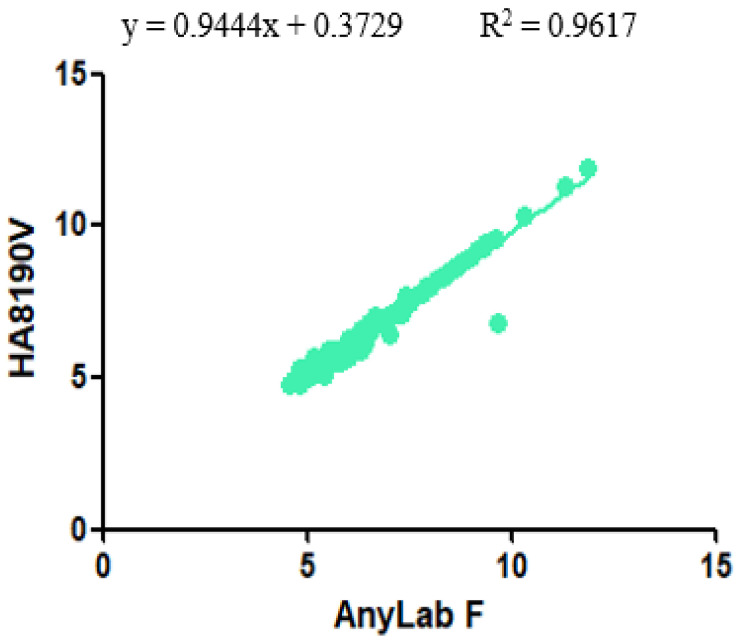
Correlation between the AnyLab F1 and HA8190V assays for the measurement of HbA1c levels (*n* = 200).

**Figure 4 diagnostics-13-03449-f004:**
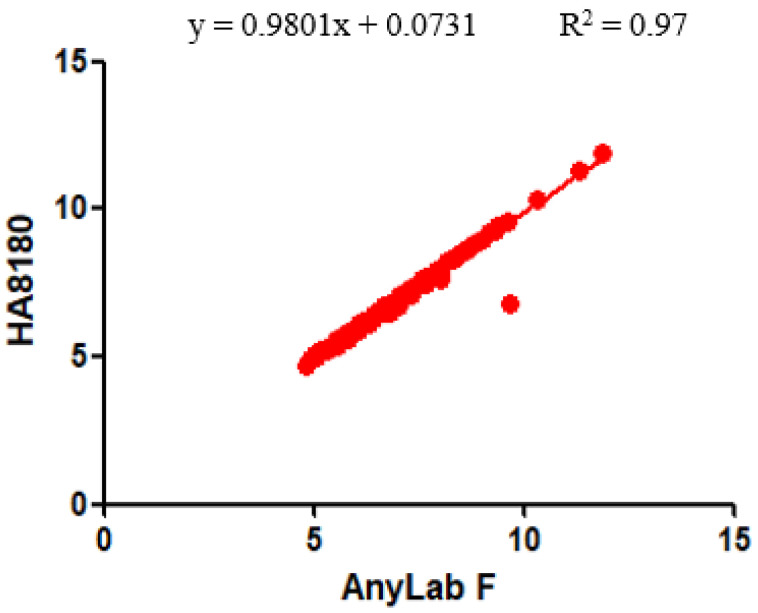
Correlation between the AnyLab F1 and HA8180 assays for the measurement of HbA1c levels (*n* = 200).

**Figure 5 diagnostics-13-03449-f005:**
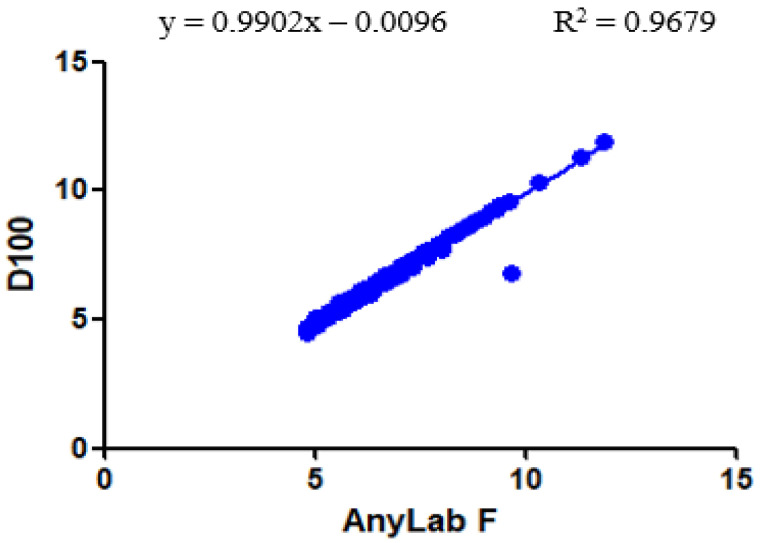
Correlation between the AnyLab F1 and D100 assays for the measurement of HbA1c levels (*n* = 200).

**Figure 6 diagnostics-13-03449-f006:**
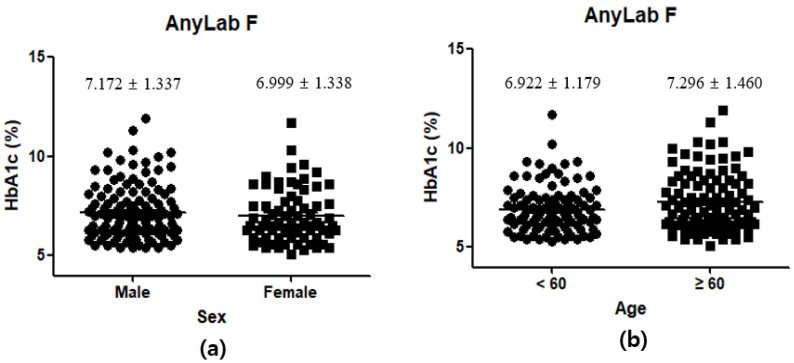
Demographic and clinical characteristics of the participants: sex (**a**); age (**b**).

**Figure 7 diagnostics-13-03449-f007:**
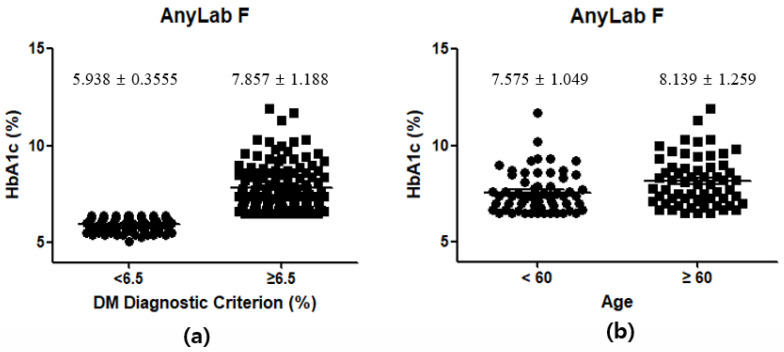
Demographic and clinical characteristics of the participants by DM diagnostic criterion (**a**) and age in DM participants (**b**).

## Data Availability

The datasets used and analyzed during the current study are available from the corresponding author upon reasonable request.

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
