# Peer review of "A Comparative Evaluation of HbA1c Measurement Methods and Their Implications for Diabetes Management"

_diagnostics, 2023, doi:10.3390/diagnostics13223449_

Round 1

Reviewer 1 Report

Comments and Suggestions for Authors

Comments:

The authors present a comparative study of HbA1c measurement methods and their implications on diabetes management. The paper is technically sound, however, there are considerable improvements that can be made to the sampling information and the authors are encouraged to resubmit after addressing the comments listed below:

1.       There are terms introduced in the abstract without using their full form such as HPLC and DM diagnosed participants. The authors are encouraged to define the terms as they first appear in the manuscript.

2.       The sampling of the study is unreliable since the ratio between the DM (Diabetes mellitus) patients to non-DM individuals is very low. The authors are encouraged to gather more information using DM diagnosed individuals.

3.       The amount of data presented in this study for the DM-diagnosed patients is very little. Authors are encouraged to gather more data before presenting their findings.

Author Response

Reviewer 1:

  • There are terms introduced in the abstract without using their full form such as HPLC and DM diagnosed participants. The authors are encouraged to define the terms as they first appear in the manuscript.

Response: Thank you for noting this oversight. In the revised manuscript, per academic writing conventions, we have ensured that these terms are independently defined in the abstract and the main text of the manuscript at their first appearance.

  • The sampling of the study is unreliable since the ratio between the DM (Diabetes mellitus) patients to non-DM individuals is very low. The authors are encouraged to gather more information using DM diagnosed individuals.

Response: Thank you for pointing out this oversight in the patient count; this has now been corrected to the accurate value: DM (122 patients) and non-DM (78 patients).

  • The amount of data presented in this study for the DM-diagnosed patients is very little. Authors are encouraged to gather more data before presenting their findings.

Response: Thank you for pointing out this oversight in the patient count; this has now been corrected to the accurate value: DM (122 patients) and non-DM (78 patients).

Reviewer 2 Report

Comments and Suggestions for Authors

The study is a critical link showing the efficacy of rapid, easy-to-use LIFAs in comparison to expensive, more difficult HPLC or other assays.

However, while the reviewer is aware of the cost and difficulties of these assays, the introduction does not clearly motivate why this correlative study is useful and why it is important.  

Therefore, the introduction needs to be significantly expanded (maybe 2-3 paragraphs) to motivate the importance and significance of performing quantitative LIFAs as opposed to currently deployed methods.  Focusing on the cost, equipment needed, etc. will highlight to the average reader the importance of enabling LIFAs.  

Also, there is CLEARLY a conflict of interest, with the first author being a part of the company that produced the diagnostic.  This needs to be addressed at the bottom of the manuscript in the appropriate section.  While this does not modify the reviewer's opinion of the quality of the work, it does need to be stated up-front and forthright.  

Also, a schematic or flow-chart showing the complexity/simplicity of each method used would greatly enhance the ability of readers to grasp the importance of the work as well as to underscore how each method obtains its output.  

Author Response

Q1. While the reviewer is aware of the cost and difficulties of these assays, the introduction does not clearly motivate why this correlative study is useful and why it is important. Therefore, the introduction needs to be significantly expanded (maybe 2-3 paragraphs) to motivate the importance and significance of performing quantitative LIFAs as opposed to currently deployed methods.  Focusing on the cost, equipment needed, etc. will highlight to the average reader the importance of enabling LIFAs. 

Response: Thank you for this comment. The introduction was expanded to describe the importance of the LIFA, as follows:

“LIFA is an emerging and increasingly significant technology in the realm of medical diagnostics [11]. In this context, the primary focus of this study was to apply LIFA to measure HbA1c levels and contrast it with conventional techniques such as reversed-phase cation-exchange chromatography and HPLC. The underlying motivation for employing LIFA in this study is rooted in the demand for diagnostic methods that are more accessible, cost-effective, and precise, particularly for monitoring conditions like diabetes [12].

Comparative studies are indispensable for determining the effectiveness and dependability of LIFA, particularly when juxtaposed with traditional methodologies. This research carries substantial importance in the sphere of medical diagnostics owing to the potential for innovation and enhanced patient care. Furthermore, this correlation study holds significance as it evaluates a novel diagnostic technology, the AnyLab F system, and gauges its performance relative to that of established approaches. The potential advantages, such as improved diagnostic precision, resource efficiency, and superior patient care, underscore the value of this study for healthcare professionals, researchers, and patients [13,14].”

Q2. There is CLEARLY a conflict of interest, with the first author being a part of the company that produced the diagnostic.  This needs to be addressed at the bottom of the manuscript in the appropriate section.  While this does not modify the reviewer's opinion of the quality of the work, it does need to be stated up-front and forthright.  

Response: Thank you for this comment. The affiliation of the first author has been revised.

Q3. A schematic or flow-chart showing the complexity/simplicity of each method used would greatly enhance the ability of readers to grasp the importance of the work as well as to underscore how each method obtains its output.  

Response: Thank you for this comment. The flowchart of the experimental design has been added. Please see the attachment.

Round 2

Reviewer 2 Report

Comments and Suggestions for Authors

The comments were not adequately addressed. 

The background still fails to explain the cost, time, infrastructure needs, and/or manpower required to execute the assays. Additionally the flow chart does not provide a stepwise protocol to execute and read the LIFA.  Finally, while the lead authors affiliation may have changed, there remains still a potential conflict of interest.  

Until these concerns are adequately addressed, the manuscript is unsuitable for publication.   

Author Response

  • The background still fails to explain the cost, time, infrastructure needs, and/or manpower required to execute the assays.
  • The explanation about the cost, time, infrastructure needs and/or manpower is added

LIFA is generally regarded as cost-effective compared to more advanced techniques like HPLC. The cost of materials and reagents for LIFA is relatively lower, making it a more budget-friendly option for diagnostic applications. On the other hand, HPLC methods typically entail higher costs due to the requirement for specialized equipment, high-quality reagents, and skilled personnel. Maintenance costs for the equipment can also be substantial [4, 6, 11].

Regarding infrastructure needs, LIFA typically does not necessitate complex infra-structure. It can be conducted in various settings, including clinical laboratories, doctor's offices, or even in the field. Basic laboratory facilities and a clean working environment are generally sufficient. In contrast, HPLC methods demand sophisticated laboratory setups with specialized equipment, such as chromatography systems and high-quality detectors. Controlled laboratory conditions, including temperature and humidity control, are essential for ensuring accurate results [8, 11].

Lateral flow immunoassays are relatively straightforward to carry out and interpret. They do not require highly specialized personnel and can be performed by trained technicians or healthcare professionals. However, HPLC techniques require skilled personnel with expertise in operating and maintaining the sophisticated equipment. Analyzing the data obtained from these methods may also require the involvement of experienced analysts or scientists with a background in chromatography and related techniques [9, 12].

  • Additionally the flow chart does not provide a stepwise protocol to execute and read the LIFA.
  • The flow chart of the LIFA protocol is added

  • Finally, while the lead authors affiliation may have changed, there remains still a potential conflict of interest. Until these concerns are adequately addressed, the manuscript is unsuitable for publication.

As suggested, the affiliation of the author has been modified to eliminate potential conflicts of interest, and the author's name has been excluded from the list of contributors.

Round 3

Reviewer 2 Report

Comments and Suggestions for Authors

No further technical comments.  The introduction is much improved and more thorough, as is the method/flow-chart.

I'd recommend adding back the other author and affiliation and just simply adding in a statement at the end of the manuscript in the appropriate section highlighting the potential conflict of interest.  

But I leave that to the journal/editor's discretion.